# A Driving Power Supply for Piezoelectric Transducers Based on an Improved LC Matching Network

**DOI:** 10.3390/s23125745

**Published:** 2023-06-20

**Authors:** Ye Feng, Yang Zhao, Hao Yan, Huafeng Cai

**Affiliations:** Hubei Collaborative Innovation Center for High-Efficiency Utilization of Solar Energy, Hubei University of Technology, Wuhan 430068, China; fengyede77@163.com (Y.F.); 102010349@hbut.edu.cn (H.Y.); whgkzj@hbut.edu.cn (H.C.)

**Keywords:** ultrasonic power supply, piezoelectric transducer, improved LC matching network, frequency tracking, power regulation

## Abstract

In the ultrasonic welding system, the ultrasonic power supply drives the piezoelectric transducer to work in the resonant state to realize the conversion of electrical energy into mechanical energy. In order to obtain stable ultrasonic energy and ensure welding quality, this paper designs a driving power supply based on an improved LC matching network with two functions, frequency tracking and power regulation. First, in order to analyze the dynamic branch of the piezoelectric transducer, we propose an improved LC matching network, in which three voltage RMS values are used to analyze the dynamic branch and discriminate the series resonant frequency. Further, the driving power system is designed using the three RMS voltage values as feedback. A fuzzy control method is used for frequency tracking. The double closed-loop control method of the power outer loop and the current inner loop is used for power regulation. Through MATLAB software simulation and experimental testing, it is verified that the power supply can effectively track the series resonant frequency and control the power while being continuously adjustable. This study has promising applications in ultrasonic welding technology with complex loads.

## 1. Introduction

Ultrasonic welding technology has the advantages of fast, environmentally friendly, safe, and reliable welding [1,2], and is widely used in industrial production [3,4,5]. In ultrasonic welding systems, the piezoelectric transducer (PT) converts the electrical energy provided by the ultrasonic power source into mechanical energy of high-frequency vibration, and the energy is transferred to the welded part to achieve welding [6,7,8]. In order to obtain stable ultrasonic energy, the ultrasonic power source requires two functions, automatic frequency tracking and power regulation [9,10]. A piezoelectric transducer is an oscillating unit that can achieve maximum power transfer only if it is made to work in a resonance state [11,12]. When the system environment and working conditions change, the piezoelectric transducer impedance and series resonance frequency also change [13,14], the frequency tracking function makes the ultrasonic power supply operating frequency follow the transducer series resonance frequency change accurately and timely to ensure that the transducer always works in the resonance state [15]. The power adjustment function regulates the electrical energy delivered to the piezoelectric transducer by the ultrasonic power supply and controls the vibration amplitude of the piezoelectric transducer output in order to adapt to a variety of welding conditions [16].

Since the dynamic branch in the equivalent circuit of the piezoelectric transducer is equated by the electromechanical characteristics [17], it is difficult for the existing technology to directly track the series resonance frequency of the dynamic branch of the transducer. Currently, the voltage and current at both ends of the transducer are often used as feedback to achieve the closed-loop control function [18]. The maximum current method uses the transducer current as the feedback quantity, and the frequency is tracked by using the characteristics of the maximum current when the transducer is working in the minimum impedance resonance [19], which is a simple method. However, the maximum current value will change with the load, resulting in a slower frequency tracking speed under sudden load changes. The phase-locked loop method uses the phase difference between the transducer voltage and current as the feedback quantity and tracks the frequency using the characteristic that the output voltage is in phase with the current when the transducer is operating at the resonance frequency [20], but the tracking bandwidth of this method is limited. Dong et al. [21] proposed a phase-locked loop improvement method with static capacitive bandwidth compensation for this problem. Zhang et al. [22] used phase and current for composite control, which can track the frequency change of the maximum vibration amplitude more accurately and quickly.

Therefore, in order to track the series resonance frequency, this paper improves the LC matching network and uses its component voltage information to discriminate the transducer series resonance frequency, and designs an ultrasonic power supply based on root-mean-square (RMS) voltage feedback control. The fuzzy control algorithm is used to track the series resonant frequency with a wide tracking range and high accuracy. The double closed-loop control method of the power outer loop and PT current inner loop is used to ensure output power stability. The structure of this paper is as follows. Section 2 analyzes the transducer equivalent model and characteristic frequency, and designs an improved LC matching network. Section 3 designs an ultrasonic driving system based on RMS voltage feedback control, and gives the frequency tracking and power control methods. Section 4 experimentally verifies the feasibility of the designed driving power supply.

## 2. Analysis of Piezoelectric Transducer and Matching Network

### 2.1. PT Equivalent Model and Characteristic Frequencies

In order to analyze the electrical behavior characteristics and characteristic frequencies of the piezoelectric transducer, the PT can be modeled by the Butterworth–Van Dyke (BVD) circuit model [23] in Figure 1. The equivalent circuit of the PT consists of static and dynamic branches connected in parallel. *C*_0_ is the static capacitor, which is the capacitance between the piezoelectric pole plates, so it is a real electrical parameter whose value hardly changes. The dynamic branch consists of the dynamic capacitor *C*_1_, dynamic inductor *L*_1_, and dynamic resistor *R*_1_ in series. The parameters *C*_1_, *L*_1_, and *R*_1_ are the reaction equivalents of the vibration operating state, characterizing the system stiffness, oscillation quality, and losses, respectively. Therefore, factors such as temperature, pressure, and operating conditions can affect the equivalent circuit parameters of the PT.

The admittance of the PT operated at frequency *f* is
(1)$$Y_{T}(\omega )=G(\omega )+jB(\omega )=\frac{R_{1}\omega^{2}C_{1}^{2}}{{\left(1-\omega^{2}L_{1}C_{1}\right)}^{2}+R_{1}^{2}\omega^{2}C_{1}^{2}}+j\left[\frac{\omega C_{1}(1-\omega^{2}L_{1}C_{1})}{{\left(1-\omega^{2}L_{1}C_{1}\right)}^{2}+R_{1}^{2}\omega^{2}C_{1}^{2}}+\omega C_{0}\right]$$
where *ω* is the operating angular frequency.

Based on Equation (1), the following equation can be obtained:(2)$${\left(G-\frac{1}{2R_{1}}\right)}^{2}+{\left(B-\omega C_{0}\right)}^{2}={\left(\frac{1}{2R_{1}}\right)}^{2}$$

The variations of conductance *G* and susceptance *B* with frequency are represented as the admittance circle shown in Figure 2. There are three characteristic frequencies [24,25]. *f*_s_ is the series resonance frequency, which is the frequency at which series resonance occurs in the dynamic branch, calculated as
(3)$$f_{s}=\frac{1}{2\pi \sqrt{L_{1}C_{1}}}$$

When the piezoelectric transducer is operated at *f*_s_, its conductance is maximum and the energy conversion efficiency is highest. Therefore, in this paper, the series resonance frequency *f*_s_ should be the target point for frequency tracking. *f*_m_ is the characteristic frequency corresponding to the maximum admittance, and the current is maximum when the PT is excited at *f*_m_. The maximum current method uses this characteristic for frequency tracking, and the target frequency is *f*_m_. *f*_r_ is the characteristic frequency at which the susceptance is zero. The voltage and current are in phase when the PT operates at *f*_r_, which is the target frequency of the phase-locked loop method. The value of 2*πf*_s_*C*_0_ is generally small, so the three eigenfrequencies are approximately equal [26].

In this paper, we use a 20 kHz PT as the research object, and the parameters of the PT under static conditions measured by an impedance analyzer are shown in Table 1. The equivalent circuit parameters are used to build the PT in a later paper.

### 2.2. Matching Network

The matching network is connected between the driving power supply and the PT, which plays a key role in the efficient and stable operation of the ultrasonic system. As mentioned earlier, The PT operated at the series resonance frequency is resistive and capacitive. To improve the transmission efficiency, a matching network needs to be designed to compensate for the capacitance, which is called tuning. The inductor–capacitor (LC) matching network is commonly used and is shown in Figure 3, where *L*_2_ is the matching inductor and *C*_2_ is the matching capacitor [27,28,29]. *C*_2_ can make the equivalent resistance lower to play the role of resistance regulation. Moreover, the LC matching network can filter the harmonic components of the ultrasonic power supply output and improve the quality of the voltage waveform applied to the PT.

In order to indirectly analyze the dynamic branch through the matching network, the conventional LC matching circuit is improved, and the improved LC matching circuit is shown in Figure 4, where *L*_m_ is the matching inductor and *C*_a_, *C*_b_, and *C*_n_ are the matching capacitors. The values of the matching network parameters are designed according to the PT operating at the series resonance frequency.

The overall equivalent impedance at series resonance frequency *f*_s_ is calculated from Figure 4. The real part indicates the equivalent resistance and the calculation formula is
(4)$$R=\frac{R_{1}C_{n}^{2}}{\omega^{2}R_{1}^{2}{\left(C_{0}C_{m}+C_{0}C_{n}+C_{m}C_{n}\right)}^{2}+{\left(C_{m}+C_{n}\right)}^{2}}$$
where *C*_m_ is the equivalent capacitance of *C*_a_ and *C*_b_ in series and *C*_m_ = *C*_a_*C*_b_/(*C*_a_ + *C*_b_).

The matching capacitance parameters *C*_m_, *C*_a_, *C*_b_ are designed according to the variable resistance requirement. Then, let the imaginary part of the equivalent impedance be 0 to achieve tuning, and the matching inductance *L*_m_ is designed as Equation (5).
(5)$$L_{m}=\frac{\omega^{2}R_{1}^{2}(C_{0}C_{m}+C_{0}C_{n}+C_{m}C_{n})(C_{0}+C_{n})+(C_{m}+C_{n})}{\omega^{4}R_{1}^{2}{\left(C_{0}C_{m}+C_{0}C_{n}+C_{m}C_{n}\right)}^{2}+\omega^{2}{\left(C_{m}+C_{n}\right)}^{2}}$$

The high quality of the voltage and current waveform applied to the PT is required. So it is verified that the filtering requirements are met after determining the matching parameters. Therefore, this paper designs improved LC matching network parameters: *L*_m_ = 0.28 mH, *C*_n_ = 440 nF, *C*_a_ = 34.375 nF, and *C*_b_ = 440 nF.

## 3. The Driving Power Supply Design

### 3.1. System Composition

Figure 5 shows the overall structure of the driving power supply, consisting of two parts: the power circuit and the control circuit. The power circuit adopts AC-DC-AC conversion technology to convert the frequency-adjustable high-frequency AC signal to drive the PT. The control circuit samples the three voltage RMS values on the modified LC matching network as the feedback signals. The output PWM signal is adjusted by the DSP controller to control the on/off of the IGBT of the full-bridge inverter circuit, thus realizing the frequency tracking and power regulation functions.

The power circuit topology is shown in Figure 6. The uncontrolled rectifier circuit composed of diodes D1–D4 converts the industrial frequency AC power to DC power and smooths the output after the *LC* filter circuit. The full-bridge inverter circuit with four IGBTs as switching devices serves to convert DC power into high-frequency AC power. In addition, the IGBTs are connected in parallel with capacitors and diodes to achieve soft switching technology, which effectively reduces the switching losses under high-frequency operating conditions [30]. The high-frequency transformer converts the inverted voltage into the power ultrasonic signal required by the transducer and acts as an electrical isolator. The improved LC matching network tuned matching makes the piezoelectric transducer work in resonance to produce ultrasonic waves.

### 3.2. Frequency Tracking

#### 3.2.1. Series Resonance Frequency Discrimination

Using the three key voltage RMS values on the improved LC matching network as feedback, it can discriminate the series resonant frequency shift of the PT, and then adjust the operating frequency of the driving power supply until it is consistent with the series resonant frequency to achieve frequency tracking. The discriminative principle is shown in Figure 7, where ***u***_1_ is the voltage on the matching capacitor *C*_b_, ***u***_2_ is the voltage on *C*_n_, and ***u***_3_ is the voltage on *C*_b_ and *C*_n_. The corresponding voltage RMS values are *U*_1_, *U*_2_, and *U*_3_, the feedback signal of this paper.

From Figure 7, Kirchhoff’s law equation is written as follows: (6)$$\left\{\begin{array}{l} u_{1}-u_{2}-u_{3}=0\\ k_{1}u_{1}-u_{2}-u_{z}=0\\ i_{0}+i_{1}-i_{n}=0 \end{array}\right.$$
where *k*_1_, the voltage division factor, is the ratio of the voltage on *C*_a_ and *C*_b_ to the voltage on *C*_b_, and *k*_1_ = (*C*_a_ + *C*_b_)/*C*_a_. From Equation (6), the vector relationship is shown in Figure 8. Here, ***u***_z_ is perpendicular to ***i***_0_ and ***u***_2_ is perpendicular to ***i***_n_. Let the angle between ***i***_n_ and ***u***_z_ be *γ*.

The three states of the PT equivalent circuit are analyzed, and the current vector relationship diagrams when the dynamic branches are resistive, inductive, and capacitive are shown in Figure 9.
(1)As shown in Figure 9a, when the dynamic branch occurs in series resonance, the dynamic branch is resistive, and ***u***_z_ is in phase with ***i***_1_, the angle between in and ***i***_1_ is equal to the angle *γ* between ***i***_n_ and ***u***_z_, then *I*_n_sin *γ* = *I*_0_;(2)As shown in Figure 9b, when the dynamic branch is inductive, ***i***_1_ lags ***u***_z_ by a certain angle, then *I*_n_sin *γ* < *I*_0_;(3)As shown in Figure 9c, when the dynamic branch is capacitive, ***i***_1_ leads ***u***_z_ by a certain angle, then *I*_n_sin *γ* > *I*_0_.

Therefore, the relationship between the current *I*_n_, *I*_0_, and sin *γ* can analyze the state of the dynamic branch of the PT. Let:(7)$$A=I_{n}\sin\gamma -I_{0}$$

The value of *A* can be calculated to determine the phase relationship between the dynamic branch voltage and current, and then determine whether the dynamic branch achieves series resonance.

The *I*_n_ = *ωC*_n_*U*_2_ and *I*_0_ = ω*C*_0_*U*_z_ are substituted into Equation (7) to obtain Equation (8).
(8)$$A=\omega C_{n}U_{2}\sin\gamma -\omega C_{0}U_{z}$$

Furthermore, let
(9)$$B=U_{2}\sin\gamma -k_{2}U_{z}$$
where, *k*_2_ = *C*_0_/*C*_n_. According to the cosine theorem, three sets of relations are obtained as in Equation (10).
(10)$$\left\{\begin{array}{l} U_{3}^{2}=U_{1}^{2}+U_{2}^{2}-2U_{1}U_{2}\cos\theta \\ U_{z}^{2}=k_{1}^{2}U_{1}^{2}+U_{2}^{2}-2k_{1}U_{1}U_{2}\cos\theta \\ k_{1}^{2}U_{1}^{2}=U_{z}^{2}+U_{2}^{2}-2U_{z}U_{2}\cos(90{}^{\circ}+\gamma ) \end{array}\right.$$

In addition, because cos(90°+γ)=−sinγ, it can be deduced that Equation (11).
(11)$$\left\{\begin{array}{l} \sin\gamma =\frac{k_{1}^{2}U_{1}^{2}-U_{z}^{2}-U_{2}^{2}}{2U_{z}U_{2}}\\ U_{z}^{2}=(k_{1}^{2}-k_{1})U_{1}^{2}+(1-k_{1})U_{2}^{2}+k_{1}U_{3}^{2} \end{array}\right.$$

After substituting Equation (11) into Equation (10), let the judgment value
(12)$$M=(k_{1}+2k_{1}k_{2}-2k_{1}^{2}k_{2})U_{1}^{2}+(k_{1}+2k_{1}k_{2}-2k_{2}-2)U_{2}^{2}-(k_{1}+2k_{1}k_{2})U_{3}^{2}$$

In summary, the judgment value *M* can be calculated by sampled RMS voltage *U*_1_, *U*_2_, *U*_3_, and corresponds to the three states of the PT as follows.
(1)If *M* = 0, the dynamic branch is resistive, and the PT works at a mechanical resonant frequency.(2)If *M* > 0, the dynamic branch is capacitive and the frequency should be increased.(3)If *M* < 0, the dynamic branch is inductive and the frequency should be reduced.

The ultrasonic system simulation model is built in MATLAB/Simulink platform. Taking the PT with series resonant frequency 20,007 Hz as the object, the relationship between the judgment value *M* and frequency *f* is obtained as shown in Figure 10. As can be seen from Figure 10, the judgment value *M* is 0 at the series resonant frequency *f*_s_ = 20,007 Hz, and |*M*| ≈ 2000 when the frequency differs from the series resonant frequency by 1 Hz. The resonant frequency fs can be tracked by controlling *M* within a certain range, and the control accuracy is high. In addition, near the series resonant frequency, the judgment value *M* changes quickly with frequency, while away from the series resonant frequency, the *M* changes relatively slowly.

Therefore, the frequency is adjusted according to the judgment value *M*. The positive or negative of *M* determines the direction of frequency change, and the magnitude of frequency change is determined by the magnitude of *M*. The *M* is controlled near 0 as the target that is tracked to the series resonant frequency. This method can avoid the influence of static capacitance to directly analyze the dynamic branch to achieve accurate frequency tracking, even in the case of matching misalignment or load change can still track the series resonant frequency of the PT. In addition, the sampling circuit of the three voltages adopts an identical circuit structure, which effectively reduces the detection error compared with the phase-locked tracking method of voltage and current phase difference, and the frequency of the PWM wave is digitally controlled by using a DSP controller with a large frequency adjustment range.

#### 3.2.2. Fuzzy Control Algorithm

Fuzzy control is based on fuzzy set theory, fuzzy linguistic variables, and fuzzy logic reasoning, which does not require an accurate mathematical model of the controlled object and has strong adaptability to nonlinear and unstable control objects. Based on the above analysis, this paper adopts fuzzy control for frequency tracking. A two-dimensional fuzzy controller with *E*, *EC*, and *DF* as fuzzy variables is used. The input variables of the fuzzy control are the judgment value *E* = *M* and the relative frequency change rate *EC* = ∆*M*/∆*f* of the judgment value *M*. The output variable is the frequency change *DF*. The block diagram of the fuzzy control system is shown in Figure 11.

In the fuzzy controller, the affiliation function adopts the trigonometric function, and the area center method is used for clarification. The actual input and output quantities are fuzzified as:(1)The basic domain of the judgment value is [−45,000, 45,000], and the fuzzy domain after fuzzification is [−6, −4, −2, 0, 2, 4, 6]. The fuzzy set composed of fuzzy language uses 7 levels (NB, NM, NS, ZO, PS, PM, PB), and the quantization factor is *K_E_* = 1/7500.(2)The basic domain of the rate of change of the judgment value is [−1800, 1800], and the fuzzy domain after fuzzification is taken as [−6, −4, −2, 0, 2, 4, 6], and the fuzzy set composed by the fuzzy language adopts 7 levels (NB, NM, NS, ZO, PS, PM, PB), and the quantization factor is *K_EC_* = 1/300.(3)The basic domain of the frequency change quantity is [−18, 18], the fuzzy domain is taken as [−6, −4, −2, 0, 2, 4, 6], the fuzzy set is divided into 7 classes (NB, NM, NS, ZO, PS, PM, PB), and the scaling factor *K_DF_* = 3.

The setting of fuzzy control rules determines the output of fuzzy control. When *M* > 0, the output of the change in frequency is positive, the frequency should be adjusted in the direction of increase, when *M* < 0, the output of the change in frequency is negative, the frequency should be adjusted in the direction of decrease, the positive and negative of *E* determines the positive and negative of ∆*f*. When the frequency is adjusted as shown in Figure 12a, the frequency is less than the series resonant frequency, the frequency is adjusted in the direction of increasing, *M*_n_ is the current moment judgment value, *M*_n−1_ is the previous moment judgment value, *M*_n_ − *M*_n−1_ > 0, then *EC* > 0, at this time, the frequency is far from the series resonant frequency, the amount of frequency change can take a larger value to speed up the search speed. When the frequency adjustment is as shown in Figure 12b, *M*_n_ − *M*_n−1_ < 0, that is, *EC* < 0, at this time the frequency is near the series resonant frequency, the frequency change amount should take a smaller value.

According to the logic judgment, if *E* and *EC* are positive, that is, the left half of the rising part of the relationship curve, the judgment value is positive at this time; the working frequency is much smaller than the series resonant frequency; the frequency change should be in the direction of increasing; and the output value should be large, expressed by the fuzzy rules as follows: if *E* is PB and *EC* is PS, then *DF* is PB. The fuzzy control rules are summarized in Table 2.

The output surface of fuzzy reasoning is shown in Figure 13. In practical applications, the fuzzy control table is generated offline. The process of frequency change is to first calculate the judgment value *M* and the amount of change in the judgment value *M* relative to the frequency based on the sampled voltage, multiply by the quantization factor to get the fuzzy input variables, and then check the fuzzy control table to get the corresponding fuzzy output variables and multiply by the scaling factor for the actual frequency change.

The simulation is built in MATLAB to verify the frequency tracking method based on the fuzzy control algorithm. A frequency less than the series resonant frequency is selected as the initial frequency in the simulation, and the frequency is adjusted from the initial frequency. Table 3 shows the three feedback voltage RMS values, judgment value *M*, and frequency regulation size of the frequency regulation process. Figure 14 shows the frequency tracking results.

After 5 frequency adjustments by the fuzzy controller, the judgment value *M* decreases, the frequency adjustment size starts to decrease, and the frequency approaches the series resonant frequency. After 8 frequency adjustments, the frequency is basically stabilized at the series resonant frequency of 20,007 Hz with an accuracy of 1 Hz, which achieves the rapidity and accuracy of frequency tracking.

### 3.3. Power Regulation

The phase shift pulse width modulation (PS-PWM) method is to regulate the duty cycle of the output voltage and the output power by changing the phase difference between the conduction of the two bridge arms of the full-bridge inverter circuit, i.e., the phase shift angle *φ* [30,31]. The IGBT driving waveforms and output voltage waveform of the PS-PWM method are shown in Figure 15, where *U*_d_ is the output voltage amplitude.

Fourier expansion of the inverter output voltage *u*_ab_ is shown in Equation (13).
(13)$$u_{\mathrm{ab}}=\sum_{n=1,3,5\cdot \cdot \cdot }^{\infty }\frac{4U_{d}}{n\pi }\cos\frac{n\varphi }{2}\sin\left[n(\omega t+\frac{\varphi }{2})\right]$$
where, *n* is the number of harmonics. As the transducer creates a high impedance to the higher harmonic current, and the impedance to the fundamental current is very small, the fundamental current dominates, and the inverter output power can be approximated as the fundamental power. The PT works at the series resonant frequency, the load fundamental shift factor is 1, and let *R* be the load equivalent resistance, then the relationship between the output active power *P* and the phase shift angle *φ* is shown in Equation (13).
(14)$$P=\frac{8U_{d}^{2}}{\pi^{2}R}\cos^{2}\frac{\varphi }{2}$$

Therefore, controlling the phase shift angle can control the output power continuously adjustable. To achieve closed-loop control of power, the output power and transducer loop current are calculated using the three voltage RMS values on the improved LC matching network and used as feedback. The transducer loop current is calculated as Equation (14).
(15)$$I_{n}=\omega C_{n}U_{2}$$

The output power is calculated as Equation (15).
(16)$$P_{z}=k_{1}\omega C_{n}U_{1}U_{2}\sin\theta$$
where,
(17)$$\left\{\begin{array}{l} \sin\theta =\sqrt{1-\cos^{2}\theta }\\ \cos\theta =\frac{U_{1}^{2}+U_{2}^{2}-U_{3}^{2}}{2U_{1}U_{2}} \end{array}\right.$$

The output power is used as the outer loop feedback, the transducer loop current is used as the inner loop feedback, and the controller is a Proportional Integration (PI) controller, the control block diagram is shown in Figure 16. The power is given according to the demand, and the deviation value is obtained after comparing it with the feedback value. The output of the outer-loop controller is used as the given value of the inner-loop controller, and then the inner-loop controller controls the phase shift angle output, so as to realize the power regulation.

The simulation is built in MATLAB to verify the power regulation of the dual closed-loop control. The power regulation process is shown in Figure 17 after 0.2 s of sudden change in the given power value from 500 W to 1000 W. After 0.02 s of regulation time, the power and current re-stabilize, and the power stabilizes at the given value of 1000 W and the current stabilizes at 3.5 A. The power and current overshoot during the regulation process is 7.8% and 1.7%, both of which are small. The simulation results verify that the dual closed-loop control has good stability.

## 4. Experimental Verification

### 4.1. Experimental Setups

Figure 18 shows the ultrasonic system experimental platform. The power circuit of the driving power supply, improved LC matching circuit, and voltage sampling circuit are integrated into the main circuit board. In the control board, dsPIC33EP128GS706 is used as the main control chip to realize the functions of AD conversion, control algorithm, and PWM generation; the RMS measurement circuit is used to process the sampled voltage signal; the IED020I12-F2 chip is used to drive the IGBT in the full-bridge inverter circuit. The transducer parameters are the same as in Table 1. The touch screen realizes human–machine interaction, online display, and setting of driving power parameters. Experiments were conducted under low voltage conditions.

### 4.2. Frequency Tracking Verification

The maximum frequency shift due to its impedance variation is approximately +/−300 Hz under different operating conditions. Therefore, the frequency tracking range of the drive power supply is designed from 19 kHz to 21 kHz. The frequency of the PWM module Auxiliary Clock (ACLK) of the dsPIC33EP128GS706 chip used in the control system is 119.632 MHz, and the PWM resolution is 1.04 ns, so the frequency accuracy can reach 0.44 Hz.

Frequency tracking frequency function based on fuzzy control algorithm is experimented with. The initial frequency is selected as the minimum frequency, i.e., 19 kHz. The current and voltage signals of the transducer are detected by an oscilloscope. Figure 19a shows the current and voltage waveforms in the detuned state of the PT, and the current and voltage have phase differences. After 0.6 s, the frequency tracking is locked, and the PT voltage and current phases reach the same, as shown in Figure 19b. The results show the effectiveness of the algorithm in accurately tracking the series resonant frequency.

### 4.3. Power Regulation Verification

Figure 20a,b show the waveforms of full-bridge inverter output voltage and PT voltage when the PT works in a resonance state and the phase shift angle *φ* is 120° and 90°, respectively. As the phase shift angle decreases, the inverter output voltage duty cycle increases, and the voltage at both ends of the transducer increases, which shows that the power can be continuously adjusted by changing the phase shift angle.

Set the given frequency parameter to 100 W by touch screen, test time 3 s. The voltage and current signals of the transducer are detected by the oscilloscope, and the power regulation process is shown in Figure 21a. Without obvious sudden changes in the voltage and current, the transducer voltage and current reached stability in about 800 ms. After stabilization, the voltage and current waveforms of the PT are shown in Figure 21b.

## 5. Conclusions

In order to make the PT convert energy efficiently and stably, we proposed a driving power supply based on an improved LC matching network. It can be summarized as follows:1.To address the problem that it is difficult to analyze the dynamic branch of a PT because its equivalent circuit has electromechanical characteristics, we designed an improved LC matching circuit. The voltage information in the LC matching circuit was used to determine the series resonant frequency of the PT. The theoretical analysis results show that it can be achieved to analyze the dynamic branch of the PT indirectly and accurately.2.The driving power supply system was designed with three voltage RMS values in a modified LC matching network as feedback. Based on the analysis of the relationship between the judgment value and frequency, a frequency-tracking method based on fuzzy control was proposed. Simulation and experiment verified that the method can effectively track the series resonant frequency with high tracking accuracy.3.The principle of PS-PWM power regulation of the full-bridge inverter circuit in the main circuit of the driving power supply was analyzed. The power and current were calculated from the three RMS voltage values of the improved LC matching network. The power control strategy of the power outer loop and circuit current inner loop was proposed. Simulations and experiments verified the performance of PS-PWM power regulation and the stability and rapidity of the double closed-loop control algorithm.

Therefore, the method proposed in this paper solves the problem that dynamic branches are difficult to analyze. The designed driving power supply is able to operate the PT in resonance and control the output power while being continuously adjustable. It is considered to be applicable in ultrasonic welding systems with large load variations. In the future, the relationship between frequency regulation and power regulation will be further studied to speed up the regulation.

## Figures and Tables

**Figure 1 sensors-23-05745-f001:**
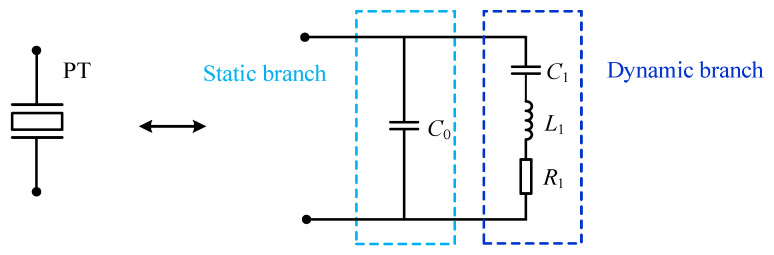
The BVD model of a PT.

**Figure 2 sensors-23-05745-f002:**
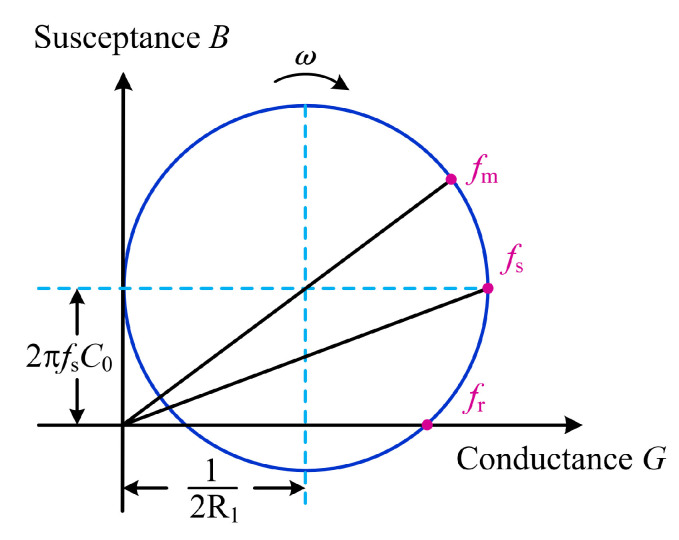
PT admittance circle.

**Figure 3 sensors-23-05745-f003:**
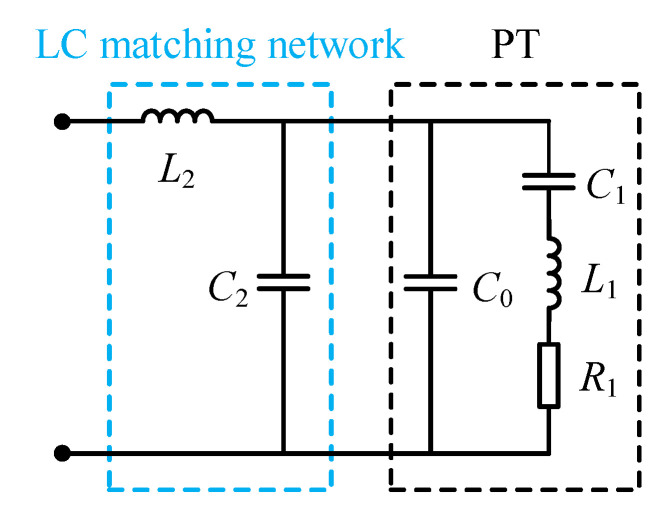
LC matching network.

**Figure 4 sensors-23-05745-f004:**
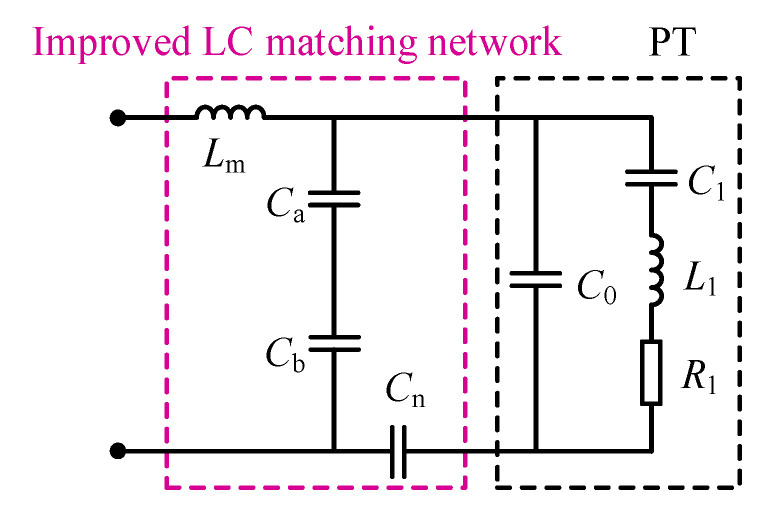
Improved LC matching network.

**Figure 5 sensors-23-05745-f005:**
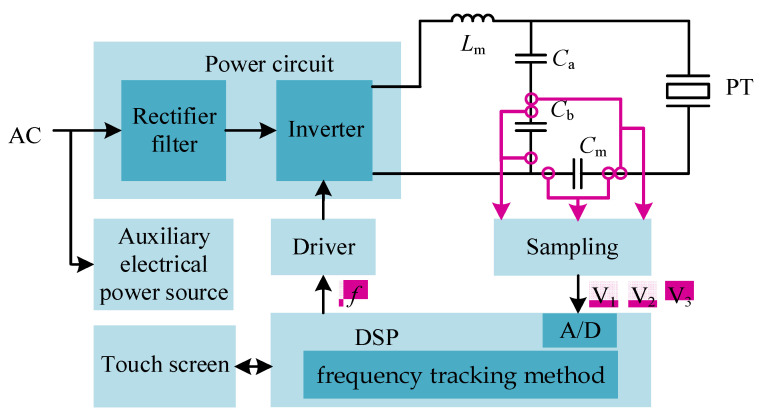
The schematic diagram of the ultrasonic driving system.

**Figure 6 sensors-23-05745-f006:**
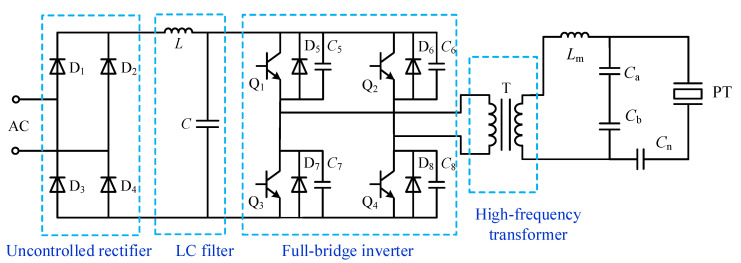
The power circuit of the ultrasonic driving system.

**Figure 7 sensors-23-05745-f007:**
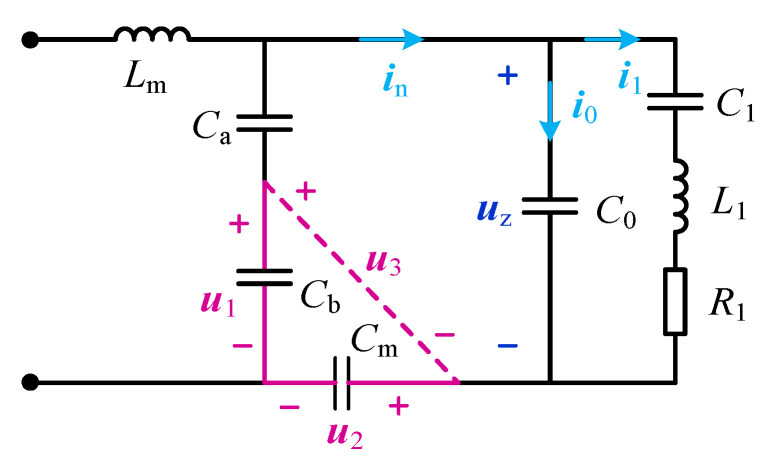
The circuit analysis.

**Figure 8 sensors-23-05745-f008:**
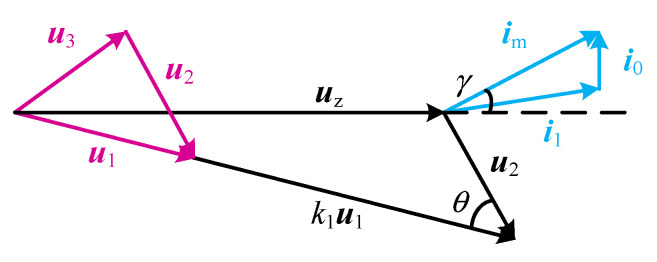
Vector diagram.

**Figure 9 sensors-23-05745-f009:**
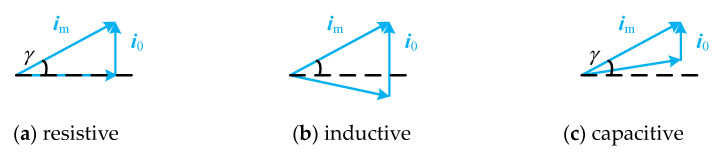
Piezoelectric transducer equivalent circuit current vector relationship diagram.

**Figure 10 sensors-23-05745-f010:**
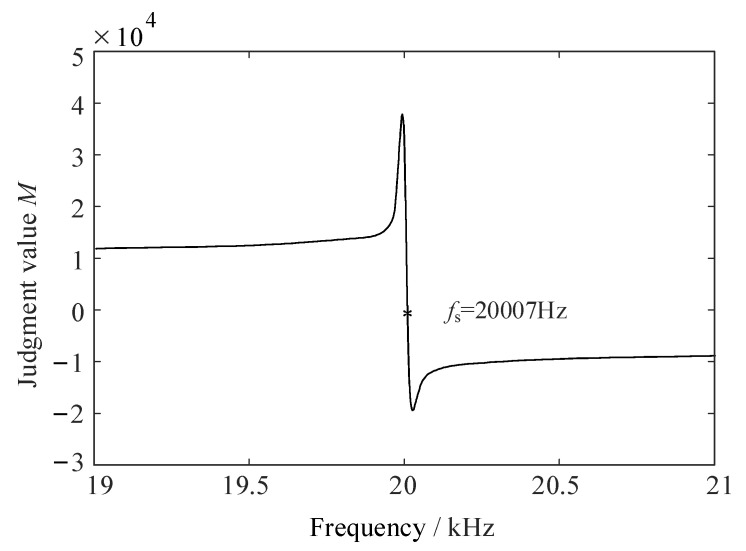
Correspondence between frequency *f* and the judgment value *M*.

**Figure 11 sensors-23-05745-f011:**
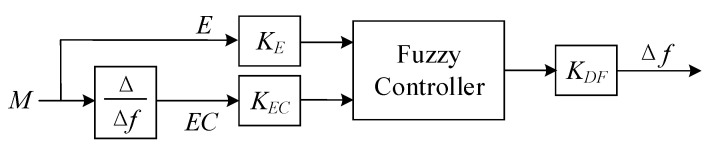
Fuzzy control system.

**Figure 12 sensors-23-05745-f012:**
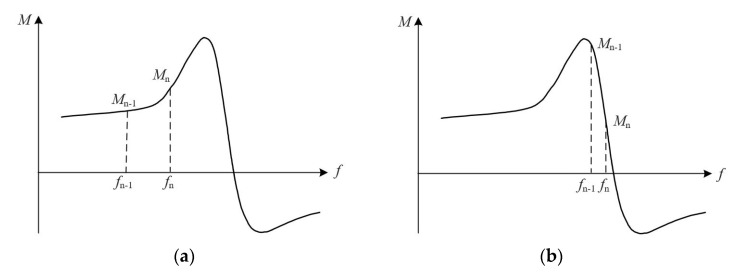
Frequency adjustment schematic. (**a**) The large frequency change corresponding to the judgment value *M*; (**b**) The small frequency change corresponding to the judgment value *M*.

**Figure 13 sensors-23-05745-f013:**
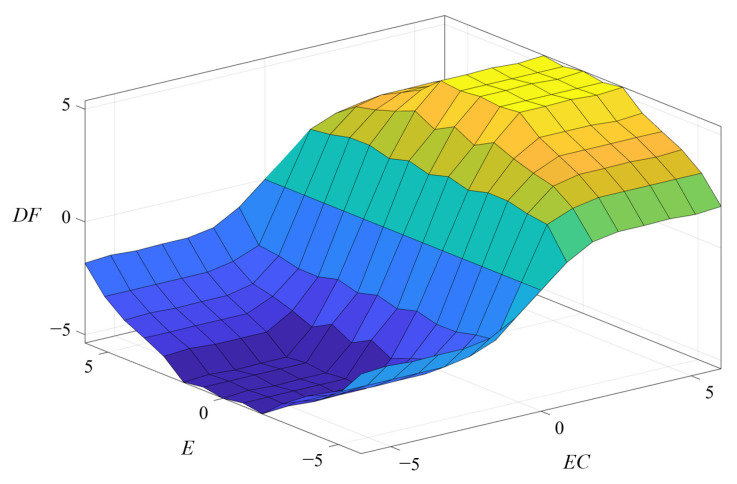
Surface description of control rules.

**Figure 14 sensors-23-05745-f014:**
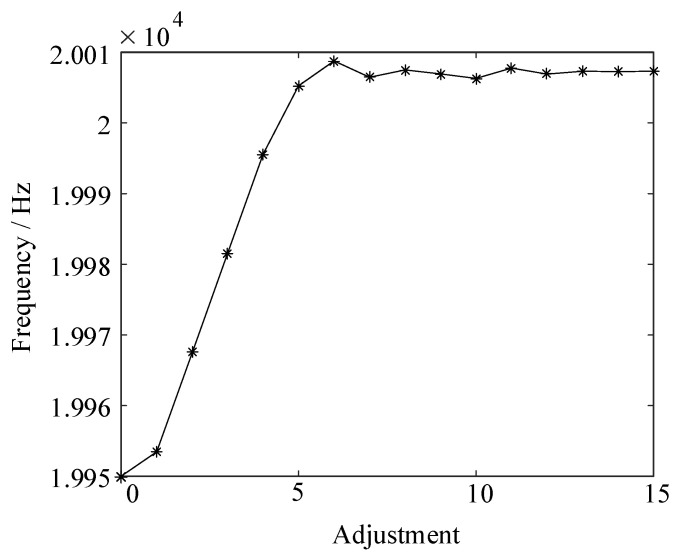
Frequency change curve of frequency tracking simulation experiment.

**Figure 15 sensors-23-05745-f015:**
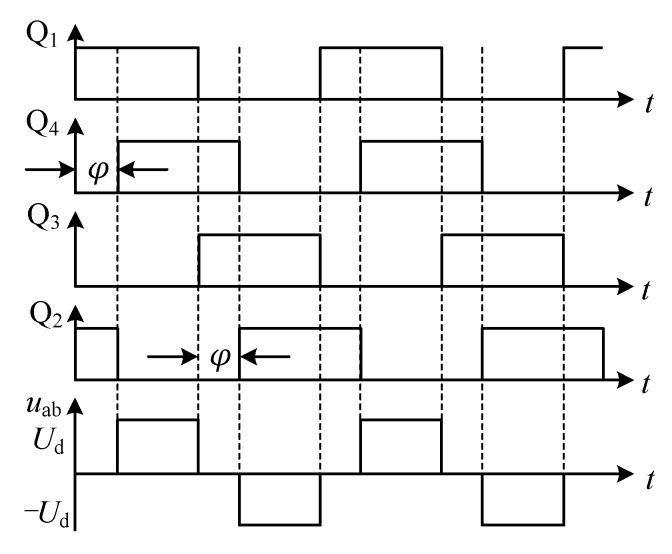
Full-bridge inverter circuit driving and output voltage waveforms.

**Figure 16 sensors-23-05745-f016:**
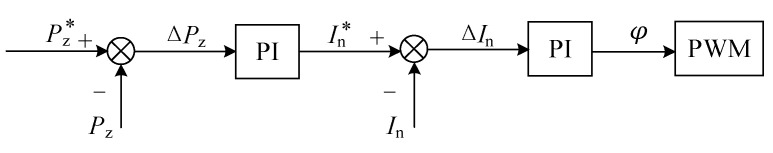
Power control block diagram.

**Figure 17 sensors-23-05745-f017:**
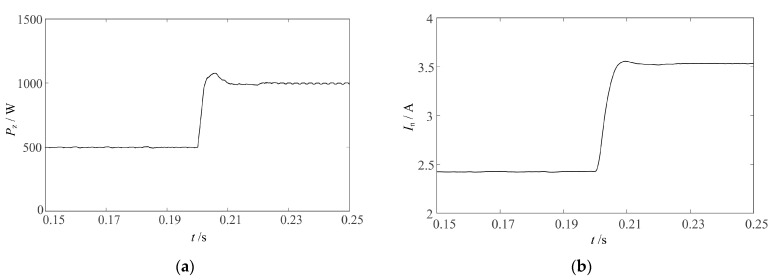
Power regulation simulation experiment results: (**a**) output power *P*_Z_ curve; (**b**) PT current RMS *I*_n_ curve.

**Figure 18 sensors-23-05745-f018:**
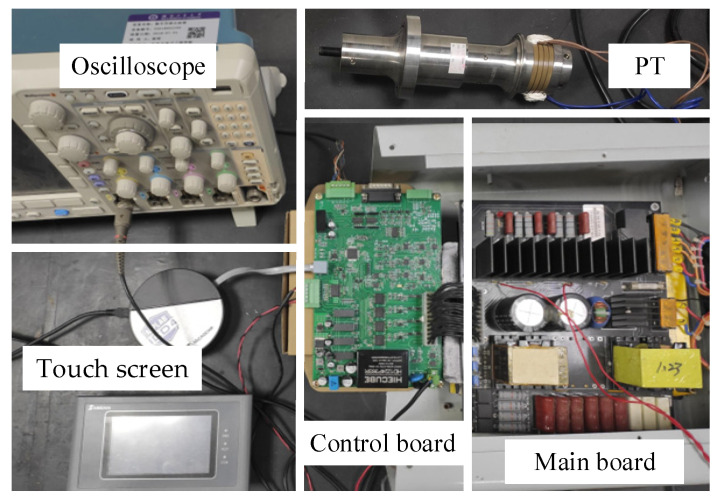
Experimental platform.

**Figure 19 sensors-23-05745-f019:**
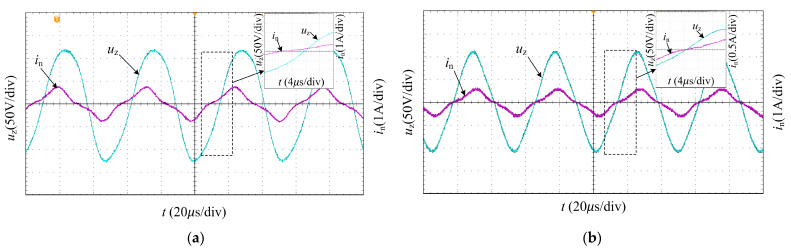
The waveforms of frequency tracking experiment: (**a**) PT voltage ***u***_z_ and current ***i***_n_ without reaching series resonant frequency; (**b**) PT voltage ***u***_z_ and current ***i***_n_ at series resonant frequency.

**Figure 20 sensors-23-05745-f020:**
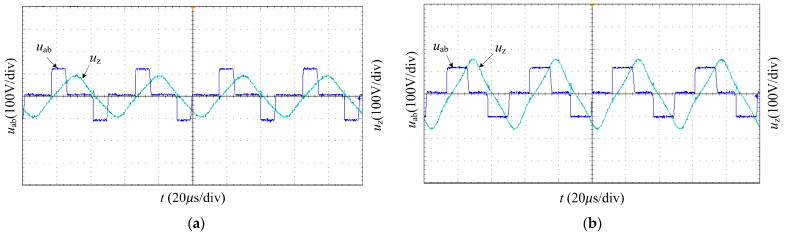
The waveforms of inverter output voltage ***u***_ab_ and PT voltage ***u***_z_ for different phase shift angles: (**a**) *φ* = 120°; (**b**) *φ* = 90°.

**Figure 21 sensors-23-05745-f021:**
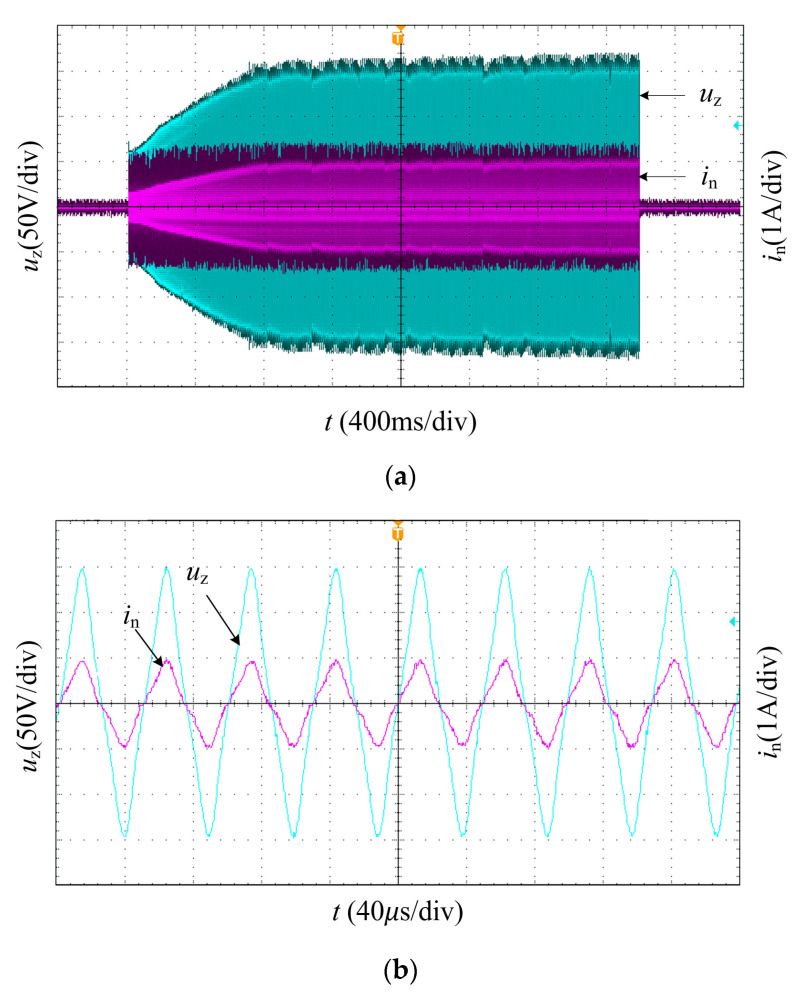
Power regulation experiment waveforms: (**a**) PT voltage ***u***_z_ and current ***i***_n_ of the power regulation process for a given power of 100 W; (**b**) PT voltage ***u***_z_ and current ***i***_n_ when the power is 100 W.

**Table 1 sensors-23-05745-t001:** Parameters of the PT.

Parameters	*C*_0_ (nF)	*C*_1_ (nF)	*L*_1_ (mH)	*R*_1_ (Ω)	*f*_s_ (Hz)
value	4.5	0.273	231.8	42	20,007

**Table 2 sensors-23-05745-t002:** Fuzzy control rules.

** *DF* **		** *E* **						
		NB	NM	NS	ZE	PS	PM	PB
** *EC* **	NB	NS	NS	NS	ZE	PS	PS	PS
	NM	NM	NM	NM	ZE	PM	PM	PM
	NS	NB	NB	NB	ZE	PB	PB	PB
	ZE	NB	NB	NB	ZE	PB	PB	PB
	PS	NB	NB	NB	ZE	PB	PB	PB
	PM	NM	NM	NM	ZE	PM	PM	PM
	PB	NS	NS	NS	ZE	PS	PS	PS

**Table 3 sensors-23-05745-t003:** Results of frequency tracking simulation experiments.

Adjustment	*U* _1_	*U* _2_	*U* _3_	*M*	∆*f*
1	37.56	32.88	13.51	20,312	3.45
2	39.08	40.25	17.43	24,935	14.10
3	40.9	49.67	27.8	28,746	14.11
4	45.12	56.81	34.26	34,437	14.02
5	38.12	53.81	43.26	18,428	9.67
6	28.78	52.77	49.81	5071	3.51
7	25.52	48.21	51.42	−2552	−2.23
8	25.3	51.18	50.9	662	1.02
9	25.23	50.02	51.14	−723	−0.57
10	25.52	50.27	50.95	−145	−0.63
11	26.21	50.78	50.93	665	1.46
12	25.21	50.03	51.22	−815	−0.84
13	26.13	50.43	50.94	291	0.38
14	25.63	50.2	51.1	−332	−0.05
15	25.98	50.46	51.09	78	0.04

## Data Availability

Not applicable.

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
