# Peer review of "A Driving Power Supply for Piezoelectric Transducers Based on an Improved LC Matching Network"

_sensors, 2023, doi:10.3390/s23125745_

Round 1

Reviewer 1 Report

It is very interest to develop piezoelectric transducer power supply with impedance matching to increase the power delivery efficiency. It will be better if the authors can do the following revisions: 

1. should the unit in be A instead of W in Figure 17?  could the Pz and In be combined such as Pz left Y and In right Y with linked  x axes? or denoted Pz (a) and In (b)?

2. In  Figure 19 ,  could make the uz left Y and in right Y with linked  x axes? add the tick labels. This suggestion also can be applied to Figures 20 and 21.

3. the figure captions for can be written more clearly with proper details.

some sentences are too long to be read. especially in the abstract and conclusion parts. could they modify them?

Author Response

  Thank you very much for your constructive comments and suggestions, I have replied and modified each of them. Please see the attachment for details.

Reviewer 2 Report

This paper designs a driving power supply based on an improved LC matching network with frequency tracking and power regulation. A fuzzy control method is used to track the series resonance frequency with wide tracking range and high accuracy. The simulated power is from 500 W to 1000 W. The simulation results verify that the dual closed-loop control has good stability. The paper is solid and here are some suggestions.

1. It is recommended to include more literature on ultrasonic welding. The welding load is different under different working conditions. Please provide the tracking range and accuracy of the designed power supply when the load changes.

2. Please compare the frequency tracking efficiency of the proposed improved LC matching network with the LC matching network.

3. In figure 19, the phase difference of the current and voltage is canceled after the frequency is locked. How to convert the phase shift to the locked frequency value in the measurement?

4. The three voltage RMS values on the modified LC matching network are used as the feedback signals. Please include them in the results.

5. Please check the grammar of line 64.

The quality of English language is acceptable.

Author Response

(The authors gave the same response as above.)

Reviewer 3 Report

Page 7 - It is not clear why the "judgement value" M is introduced.Whether the circuit is resistive, capacitive or inductive can be decided e.g. from the phase shift value.

Line 248 - It is not clear why "M" is relabelled "E". 

Figure 19 - The graph in the figure is very small, it is hard to tell if the phase shift in figure "b" is really zero. 

Author Response

(The authors gave the same response as above.)

Round 2

Reviewer 1 Report

The paper can be accepted as the current format.